# Grasping Unstructured Objects with Full Convolutional Network in Clutter

**Tengteng Zhang and Hongwei Mo ***

College of Intelligent Systems Science and Engineering, Harbin Engineering University, Harbin 150001, China; zttdouble@hrbeu.edu.cn
* Correspondence: mhonwei@163.com

**Abstract:** Grasping objects in cluttered environments remains a significant challenge in robotics, particularly when dealing with novel objects that have not been previously encountered. This paper proposes a novel approach to address the problem of robustly learning object grasping in cluttered scenes, focusing on scenarios where the objects are unstructured and randomly placed. We present a unique Deep Q-learning (DQN) framework combined with a full convolutional network suitable for the end-to-end grasping of multiple adhesive objects in a cluttered environment. Our method combines the depth information of objects with reinforcement learning to obtain an adaptive grasping strategy to enable a robot to learn and generalize grasping skills for novel objects in the real world. The experimental results demonstrate that our method significantly improves the grasping performance on novel objects compared to conventional grasping techniques. Our system demonstrates remarkable adaptability and robustness in cluttered scenes, effectively grasping a diverse array of objects that were previously unseen. This research contributes to the advancement of robotics with potential applications, including, but not limited to, redundant manipulators, dual-arm robots, continuum robots, and soft robots.

**Keywords:** deep reinforcement learning; full convolutional network; robotics; unstructured objects; dexterous grasp

## 1. Introduction

The literature presents numerous works on vision-based robot dexterous manipulation [1], especially focusing on end-to-end deep learning-based schemes that significantly improve grasping accuracy and efficiency [2,3]. Indeed, a deep convolution neural network can learn the complicated feature representation from a huge quantity of data instead of relying on handcrafted feature extraction. Thus, many researchers have attempted to train a robot to grasp objects utilizing deep learning-based solutions, primarily by estimating the robot's operation pose, which can be broken down into two categories. One scenario involves segmenting and identifying the objects before estimating the operation pose by employing a well-known object model to address the grasping task in complicated scenes [4]. Another case is that the end-to-end operation pose was determined utilizing image or point cloud data [5]. Although the latter category affords a better generalization ability, the cost of training data prevents its widespread industrial application. Nevertheless, advanced deep learning technology allows deep convolution neural networks to perform exceptionally well extracting features from images.

In reality, grasping enables robots to manipulate and interact with objects, performing tasks, such as moving, placing, and assembling. Through autonomous exploration and learning, robots enhance their understanding of the environment and objects. This allows them to engage in closer human interaction, such as delivering objects and collaborating on various tasks. For instance, Google brain employs QT-opt [6], a reinforcement learning-based extensible self-supervised learning approach. Furthermore, multiple robots

are utilized simultaneously to train the robot for a grasping task, aiming to shorten the training time and addressing that deep reinforcement learning involves more training steps. Through sparse rewards [7], Breyel et al. ensured that the robot developed a superior grasping method. Nevertheless, a continuous (non-sparse) reward function causes local optimization to be problematic, and, therefore, Berkeley developed a reward guidance algorithm to address this issue [8].

Currently, most robotic grasping systems operate in a structured environment and suffer from poor robustness and flexibility, requiring reset whenever the environment, the grasping task, or the object's condition change. However, when the objects are irregular or stacked, the grasping task difficulty increases significantly. To compensate for that, current studies typically derive image data from visual sensors and extract features manually, aiming through supervised data and conventional machine learning algorithms to determine the corresponding relationship between these artificially designed features and the robot's grasping pose [9].

To date, various DRL algorithms have been proposed for manipulation intelligence and autonomous learning. Given that the manipulator and the complex dynamic environment develop an interactive relation, the robot can learn how to grasp independently due to the evolvement of reinforcement learning strategies, opening up the possibility of autonomous control. Thus, this research develops a novel deep reinforcement learning-based approach for the intelligent grasping control of robots. Figure 1 depicts the proposed system's general layout, where two RGB-D cameras extract images and combine image data to create visual perception in simulation. Two fully convolutional action neural networks choose the best action based on these features.

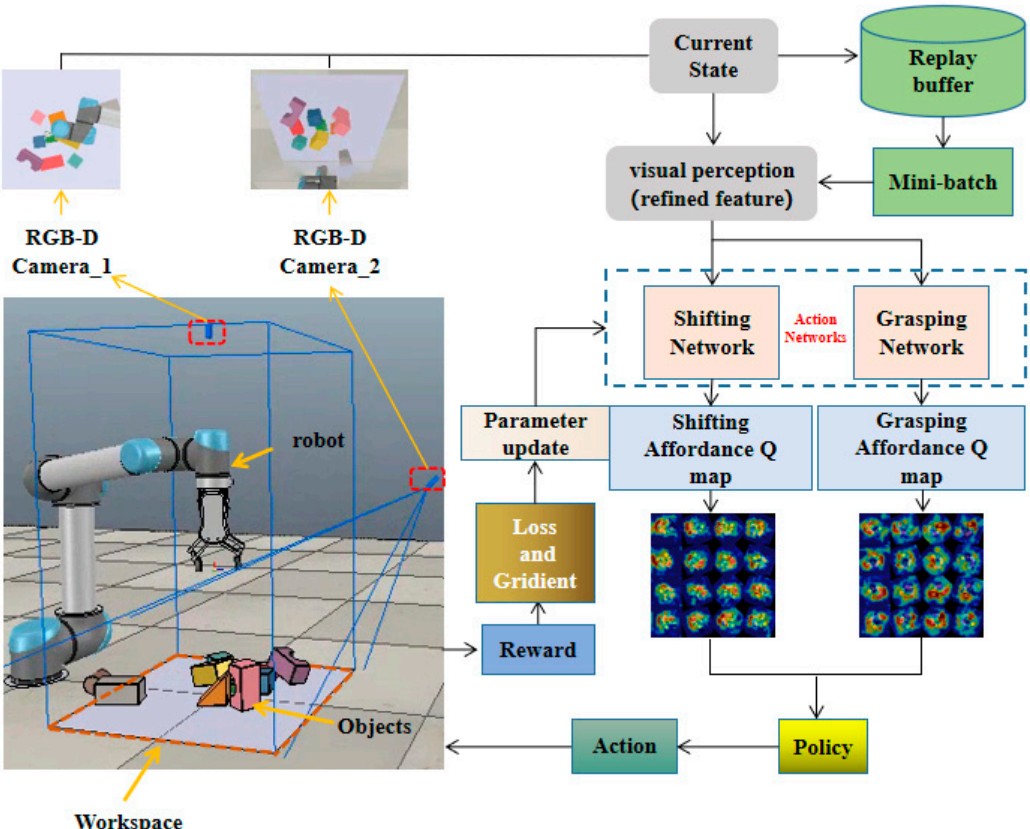

**Figure 1.** Overview of the proposed system.

In summary, this paper's contributions are as follows:

(1) Unstructured environment: We present a unique Deep Q-learning (DQN) framework combined with a full convolutional network suitable for end-to-end grasping of multiple adhesive objects in a cluttered environment.

(2) Smaller training data: To improve data utilization, we perform mini-batch training instead of training on large amounts of data by storing the transitions in the replay buffer.

(3) Reduced training time and remarkable results: We evaluate the proposed method in real-world experiments. For cluttered situations, the experimental results indicate the grasping rate of invisible objects is up to 88.9% with $282 \pm 3$ picks per hour.

The remainder of this paper is organized as follows. Section 2 briefly presents the related work. Section 3 introduces the proposed Attention-DQN (ADQN) algorithm, and Section 4 presents the experimental results and learning process. Finally, Section 5 concludes this work.

## 2. Related Work

The studies of robot grasping that are most pertinent to our research are mentioned below. The application of robot manipulation for grasping unseen objects has gained a lot of traction.

### 2.1. Vision-Based Grasping

Learning and control in robotic systems aided with neural networks, which could achieve better performances in terms of efficiency in manipulation. In recent years, visual-based perception has been widely used for robotic grasping tasks, where object features are extracted to guide the decision-making process. For instance, Lenz et al. [10] suggested that a good five-dimensional grasp representation can be back-projected into a seven-dimensional grasp representation, assisting a robot in grasping. Most previous works focus on single or specific objects in structured environments, requiring a prior knowledge of the target object and exploiting a corresponding feasible grasping set [11]. For the scenario where the objects are invisible or similar, most researchers exploit information, such as shape, color, and texture [12]. However, this strategy is inefficient since it is impossible to create a database of known objects or learn the discriminant equation for grasping without a given model and prior experience. Additionally, shape recovery or feature extraction commonly relies on signals derived from sensors, which are then exploited heuristically to generate a feasible grasp.

### 2.2. Attention Mechanism and Deep Q Network

Robotic grasping in cluttered environments is a challenging problem that has garnered significant attention in the robotics community. Researchers have explored various approaches to tackle this problem, employing both analytical and learning-based techniques [13–15]. The attention mechanism efficiently extracts high-quality elements from extensive information with limited resources [16], first applied to image classification by Mnih et al. [17]. The attention mechanism reduces temporal complexity and has already achieved remarkable results in various types of deep learning tasks, such as natural language processing, image recognition, and voice recognition. Moreover, it has also demonstrated inspiring performance in reinforcement learning projects [18–20].

At present, many applications are based on Q-learning to improve robot performance. Due to the large space occupied by Q-learning Q-table, it cannot solve some problems in the high-dimensional state space [21]. Therefore, the DQN improves Q-learning by innovatively combining Q-learning with reinforcement learning and adjusting the Q-value using a neural network. The key principle of DQN is to approximate the value function, and the experience replay mechanism breaks the correlation between data. Although a standard DQN method can solve MDP problems, it cannot be effectively generalized to unknown environments. Hence, Liang et al. proposed a knowledge-induced DQN (KI-DQN) algorithm to overcome the generalization challenges [22].

### 2.3. Grasping in Clutter

Many learning-based approaches have tried to overcome the problem of grasping objects from a cluster of multiple objects [2,23,24]. For instance, Kalashnikov et al. [25] introduced a scalable self-supervised vision-based reinforcement learning framework to train seven robot setups for 5.8 k grasp attempts, involving learned pre-grasping manipulation, like pushing and reactive grasping. Another approach that has been proposed is the use of deep Q-learning to learn pushing and grasping motion primitives [26], which achieved a task success rate of 85% for cluttered objects after approximately 2000 training episodes.

Unlike existing works [6,10,11] employing AlexNet and a pre-grasping dataset for feature extraction, we rely on ResNet [27]. Furthermore, the attention mechanism is employed to enhance the expressiveness of the target, enabling better adaptation and grasping actions for predicting rewards, then extracting the workspace features and producing the target affordance map after the action network. Related to our work, Deng et al. [28] recently proposed a DQN framework combining grasping and pushing. Finally, opposing many current grasping algorithms that utilize reinforcement learning, our experiments demonstrate that the developed method operates well on unseen objects in cluttered environments.

## 3. Problem Formulation and System Description

This study investigates how robots acquire shifting and grasping skills based on DRL. Figure 2 depicts the system framework in broad strokes, which can solve the problems of low efficiency, low success rate, and unsatisfactory cooperative effect and enhance the robot grasping ability in complicated environments.

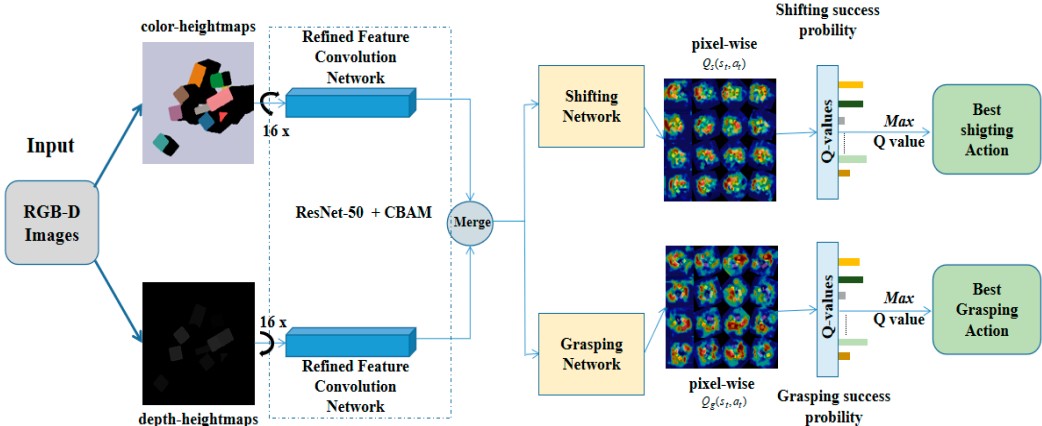

**Figure 2.** Overview of the proposed Attention Deep Q-learning (ADQN) architecture.

### 3.1. Problem Formulation

The proposed solution employs two depth cameras and a parallel-jaw gripper and investigates the combined job of shifting and gripping objects. Specifically, the robot executes an action $a_t$ under the current policy $\pi$ in the state $s_t$ of time $t$, then moves to the next state $s_{t+1}$ and receives the immediate reward $r_{t+1}$ with the action–state value function $Q_{i+1}^\pi(s_t, a_t)$ defined as:

$$Q_{i+1}^\pi(s_t, a_t) = E_{s+1}[R_{t+1} + \gamma \max Q_i^\pi(s_{t+1}, \pi(s_{t+1})] \tag{1}$$

The observation sequences are obtained by collecting camera images (state, action, reward). To maximize the discounted sum of future rewards, i.e., $G_t = \sum_{i=0}^{\infty} \gamma^i R_{i+t+1}$, the robot generally searches for a policy $\pi$. When $I \to \infty$, our objective is to hasten the Q-function

convergence in the course of a continuous environmental interaction, i.e., $Q^\pi \to Q^{\pi^*}$. Finally, the optimal action the robot performs in the state $s_t$ is derived as follows:

$$a_t^* = \underset{a_t}{\mathrm{argmax}} Q^{\pi^*}(s_t, a_t) \tag{2}$$

### 3.2. Attention-Based DQN

The attention network can elevate object perception. Thus, our attention architecture (CBAMNet) is a convolutional block attention module that is mainly based on the deep residual network (ResNet-50), with the network comprising a convolutional layer and four attention blocks. The spatial attention (SA) mechanism and channel attention (CA) mechanism are used for residual concatenation in the attention blocks. SA generates the channel attention map, which focuses on the global information, and CA focuses on the spatial feature map of the attention space and target space. The CA and SA are independent and are sequentially combined to enhance attention to the position and feature information of the objects in the workspace. The output features are merged and input into two action networks to generate the shifting and grasping action visualization maps. The pixel-wise prediction value $Q$ and the probability of the action are obtained using the greedy strategy. Moreover, self-training aims to optimize the target value:

$$Q_{i+1}(s_t, a_t) = R_{t+1}(s_t, s_{t+1}) + \gamma \underset{a}{\max} Q(s_{t+1}, a; \theta_{t+1})] \tag{3}$$

where $Q_{t+1}$ is the predicted value of the executed action, $R_{t+1}(s_t, a_t)$ is the reward value obtained after executing action $a_t$, $\theta_{t+1}$ is the network parameter at time $t + 1$, and the maximum predicted value $Q$ is derived from selecting the optimal action.

### 3.3. Reward Function

The sparse feedback is a common problem for reinforcement learning, which is fatal to convergence. In this work, the robot is rewarded only when it successfully grasps an object. The hierarchical reward is designed at the initial phase of the interactive training between the robot and the system environment.

$$r = \begin{cases} 1 & successful \quad grasping \\ 0 & unsuccessful \quad grasping \\ \frac{1}{\Delta x} - E(r) & movement \quad process \end{cases} \tag{4}$$

where $E(r)$ is the expectation of the relative reward estimation and $\Delta x$ is the distance from the center of the two-finger gripper to the center of the nearest object. The coordinates $(x, y, z)$ of the gripper and the coordinates $(x_i, y_i, z_i)$ of the object in the base coordinate system are obtained simultaneously because training is conducted in a virtual environment system. Distance $\Delta x$ is derived as follows:

$$\Delta x = \sqrt{(x - x_i)^2 + (y - y_i)^2 + (z - z_i)^2} \tag{5}$$

We use the mean reward to update the reward expectation:

$$E(r_{new}) \leftarrow \frac{1}{n}\frac{1}{\Delta x} + \frac{n-1}{n} r_{old} \tag{6}$$

The reward design is based on the motion control and grasping operations. Thus, it can be roughly inferred that the success of grasping depends on the relative position and posture of the gripper and object. The success of preliminary grasping attempts can deduce the quality of posture directly, i.e., the reward feedback of posture is proximal. The motion control process reflects the position change, and its feedback is challenging to learn. Algorithm 1 shows DQN based on CBAMNet.

---

**Algorithm 1** Proposed DQN based on CBAMNet

---

Initialize the parameter $\theta$, time step $t$ and reward $r$ of network model
Initialize an experience replay buffer $D$
Initialize action-value function Q with random weights
**for** number of episodes **do**
    Extracting intermediate features from two identical attention networks and merging
    **for** $t = 1, T$ **do**
      The behavior strategy selects the next action: $a_t = \max_a Q^*(s_t, a; \theta)$
      Execute the action and feedback the reward value from the environment: $r_t$
      Store transition $(s_t, a_t, r_t, s_{t+1})$ in $D$
      Sample random minibatch of transition $(s_i, a_i, r_i, s_{i+1})$ from D
      Calculate the target value: $y_t = R_a(s_t, s_{t+1}) + \gamma \max_a Q(s_{t+1}, a'; \theta_{t+1})$
      Calculate the loss function, optimize objective and update network parameters
      **if** $Q_s(s_t, a_t) = \max_a Q_s^*(s_t, a; \theta)$ **do**
          Execute the action of shifting
      **if** $Q_g(s_t, a_t) = \max_a Q_g^*(s_t, a; \theta)$ **do**
          Execute the action of grasping
  **if** the objects out of the training workspace **then**
          Break;
    **end for**
  **end for**

---

## 4. Experiment Results

Usually, there are multiple objects presented in the actual grasping scene, which brings significant difficulties for object grasping. We carried out model training in the simulation environment to lessen the loss of the robot. The trained model would then be transferred to the actual robot arm. We verified how to improve the grasping success rate in real-word experiments.

### 4.1. Experimental Setup

In the simulation, the proposed method is evaluated in various scenarios to exhibit its effectiveness using a modeled 6-DOF robot arm UR5 and a two-finger parallel gripper RobotIQ 2F-85 with an adjustable range of 0~85 mm. All experiments are on the v-rep3.6 simulation platform based on bullet 2.8. Moreover, an Intel RealSense D435 depth camera is used as an overhead camera, and another identical camera is deployed at 45°. Our model is trained in PyTorch with an NVIDIA-Tesla T4 on an Intel Xeon Golden CPU 2 × 6128 clocked at 2.3 GHz. Our physical experiments employ the RealSense D435 RGB-D camera and a physical robot (as shown in Figure 3). The FLEXIV rizon4 is a 7-DOF robot arm with 4 kg maximum load. The camera is attached to the end-effector, affording a good visual coverage of the graspable objects

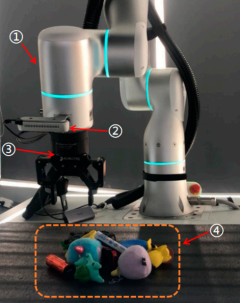

**Figure 3. Experiment in real environment**. Our robot is able to grasp unseen objects with precise perception, decision making at a high rate of speed, reduced training time, and receive remarkable results. Our experimental setup consists of ① a cooperative robot, ② a depth camera, ③ a manipulator with two-finger gripper (85 mm opening, maximum load 5 kg and 24 V DC power supply), ④ a manipulation platform of various unknown objects.

*4.2. Training*

In the simulation environment (Figure 4), multiple objects are placed randomly in a challenging arrangement within a workspace of 1 m × 1 m. To expand the data sample, the images captured by the cameras are rotated 16 times per 22.5$^\circ$ and then input into the same attention network (CBAMNet) for intermediate feature extraction and fusion. The Shiftnet and Graspnet action networks conduct full convolution training based on the fused intermediate features to obtain the pixel-wise Q value maps for action prediction. According to the maximum Q-value map, the 3D point cloud is converted from the camera coordinates to the robot coordinates to calculate the robot's contact point position ($x, y, z$) in the workspace. The environment is reset if the episode terminates or the objects are out of the training workspace. The training parameter settings are listed in Table 1.

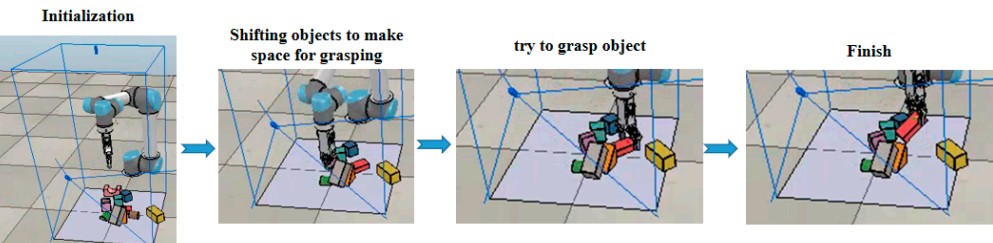

**Figure 4.** During training in the simulation environment, multiple objects are randomly generated, with their states being uncertain in each episode. The scenarios involve objects under adhesion and stacking scenarios and the shifting action is used as the grasping operation to create a better space and finally succeed in grasping.

**Table 1.** Parameter settings.

| Parameter | Value |
| :---: | :---: |
| learning rate: $\alpha$ | 0.0001 |
| exploration rate (start): $\varepsilon_i$ | 0.01 |
| exploration rate discount: $\zeta$ | 0.95 |
| discount factor: $\gamma$ | 0.9 |
| replay buffer capacity: D | 500,000 |
| mini-batch | 64 |
| max-episode | 4000 |
| Optimizer | Adam |

*4.3. Testing*

The performance of the training model is evaluated on 30 different operation scenarios (see Figure 5). Each case involves unknown objects in irregularly placed scenes, with other objects blocking the robotic tasks. The environmental state changes after performing the shifting action, providing sufficient space for the grasping action. Finally, the robot successfully grasps the object. We execute 10 runs per test, and the threshold of the action numbers is set to 2 × m per episode (m is the number of objects that must be picked per case) per episode. The performance is evaluated based on the average grasp efficiency, defined as $\dfrac{\sum_{i=1}^{runs} successful \quad numbers}{\sum_{i=1}^{runs} action \quad numbers}$, and the average grasp success rate of each object, defined as $\dfrac{\sum_{i=1}^{n} successful \quad numbers}{n(epoch \quad numbers)}$. The environment resets for the next grasping epoch if the robot grasps all objects within the threshold of action numbers and not before exceeding the threshold. For both metrics, a higher value is better.

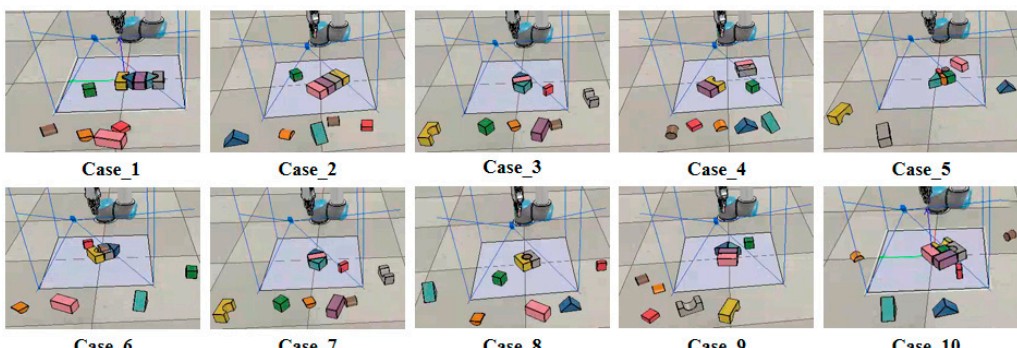

**Figure 5.** Testing cases: the robot grasps unknown objects for multi-irregular cases. During training in the simulation environment, multiple objects are randomly generated, with their states being uncertain in each episode. The scenarios involve objects under adhesion and stacking scenarios and the shifting action is used as the grasping operation to create a better space and finally succeed in grasping.

### 4.4. Comparisons with Previous Approaches

The dexterous operation skills are implemented in the v-rep simulation environment to evaluate the proposed method's effectiveness. The subsequent trials aim to prove that our method can effectively speed up training and that the action policy is effective, boosting action cooperation and grasping ability. We test our scheme against the following baseline methods: (1) Grasping only, a greedy deterministic grasping strategy [29], where the full convolution network predicts the action, and a greedy strategy selects the next action. The robot only selects the action with the maximum Q value, as defined by the greedy strategy. (2) The VPG-target maps the action Q-value through two action full convolution networks and adopts the reinforcement learning method to learn the synergy between shift and grasp [30]. (3) The vanilla Q-learning and deep vanilla Q-network (DQN) algorithms [21].

We set the maximum threshold number of actions taken during training. When the threshold is reached, we reset the environment for grasping and begin the subsequent training round. Then, a new scene will be generated for the subsequent training if the target object is successfully grasped or if there is no object in the entire training region. The robot will undergo 4000 grasping attempts, and Figure 4 compares the training curves using different techniques.

The research results (see Figure 6) indicate that, compared to the Q-learning framework [21], our proposed approach presents better completion and success rates. Additionally, even though the DQN algorithm creates endless samples in real time based on the experience replay pool for supervised training, it uses a random policy to select actions and disregards the synergy of shifting and grasping, decreasing the grasping success rate [31]. The Grasping-only method solely relies on the grasping strategy with no shifting action on the unknowable environment [29]. Moreover, the grasping performance varies from 40% to 60% and is inconsistent. Although the shifting action can alter the structure of the unknown environment based on the VPG-target approach [30], it simply encourages changing the environment's structure, and the training effectiveness is only between 50% and 65%, making it rather unsatisfactory. Although our approach performs poorly in the first 1000 iterations of training, it significantly improves in the later training stages, reaching an effectiveness of 80% and 90%, with the maximum successful grasping reaching 88.9%. The grasping success rate, grasping efficiency (per hour), mean time(s) to complete one action, number of actions, and successful objects are further analyzed and reported in Table 2. The 30 various arrangements of testing instances involve 147 objects, with our method outperforming existing methods.

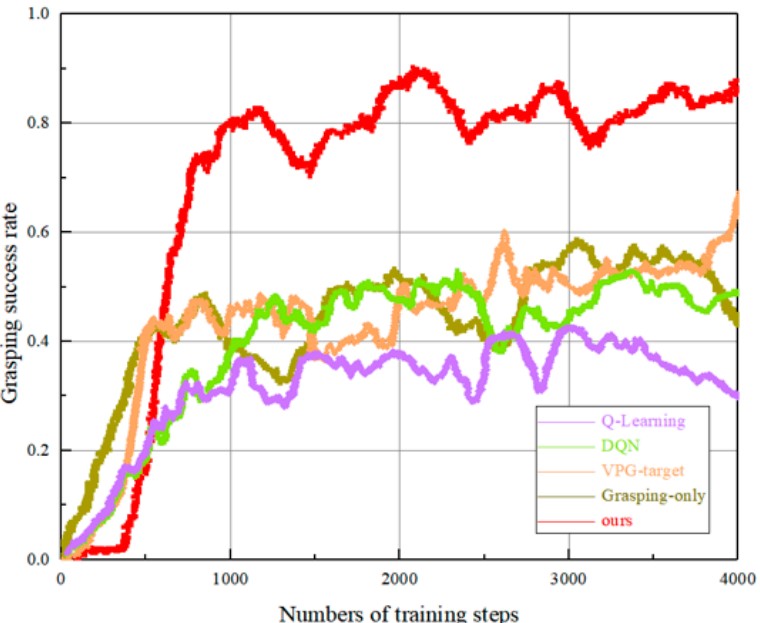

**Figure 6.** Training curve comparison between different methods.

**Table 2.** Quantitative results considering the grasp success rate on test objects.

|  | Q-Learning | DQN | Grasping-Only | VPG-Target | ADQN (Ours) |
|---|---|---|---|---|---|
| **Actions** | 8.37 k | 7.59 k | 7.41 k | 6.57 k | 5.39 k |
| **Objects** | 61 | 77 | 82 | 96 | 133 |
| **Success rate (%)** | 41.5 | 52.3 | 55.7 | 65.3 | 90.4 |
| **Mean Time (s)** | 14.6 | 13.9 | 13.4 | 12.8 | 11.5 |
| **Efficiency (per hour)** | 102 ± 5 | 135 ± 3 | 150 ± 4 | 183 ± 5 | 282 ± 3 |

### 4.5. Grasping in Clutter Scenarios

**Physical Experiments**. In each grasp attempt, our network receives the visual signals from the depth camera mounted on the robot end-effector. The color heightmaps ($4 \times 224 \times 224$) and depth heightmaps ($1 \times 224 \times 224$) are rotated 16 times and input into two identical attention networks (attention block $\times$ 4) for training. These networks comprise a $7 \times 7$ convolution and maximum-pooling layer to fuse channel attention and spatial attention (every attention block conducts $1 \times 13 \times 11 \times 1$ convolution). The fused features ($512 \times 14 \times 14$) are input into two identical action networks using a greedy strategy to generate the Q value maps of shifting and grasping actions. Then, the output is processed by two fully convolutional layers, after which the network outputs the probability of grasp success using a softmax function to guide the robot's dexterous operation. Figure 7 illustrates that our system effectively grasps unknown objects in clutter.

Finally, we evaluated whether the learned model is able to successfully grasp the unseen objects in cluttered environments. Specifically, we evaluate our method on 145 grasping attempts and reveal that the robot is successful 111 times. Table 3 illustrates the results of various methods in real-world grasping experiments. It is noticeable that the object-grasping performance is obviously poor in [24] compared with other methods [32,33]. The experimental results demonstrate that our proposed model achieves around a 91.4% grasping success rate in simulation and 88.9% in real-world execution, with fewer real data for training. This is a considerable improvement over prior studies seeking to translate simulation to reality.

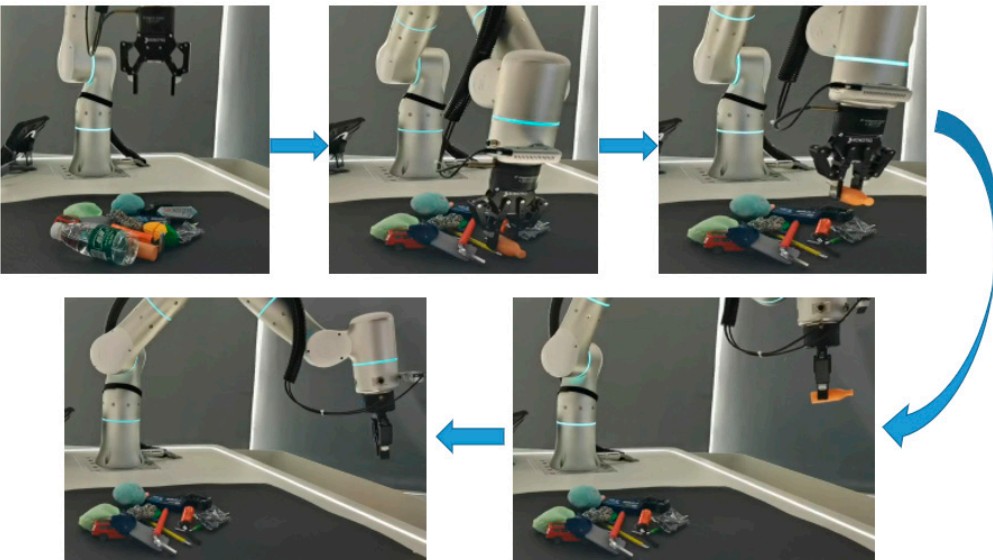

**Figure 7.** Screenshots of grasping in clutter. Our system is capable of planning to grasp the object.

**Table 3.** The results of various methods in real-world grasping experiments.

| Authors | Physical Grasping | Success Rate (%) |
|---------|-------------------|------------------|
| Morrison [32] | 126/150 | 84% |
| Pinto [24] | 97/134 | 72.3% |
| Xu [33] | 123/143 | 86% |
| ADQN (Ours) | 129/145 | 88.9% |

## 5. Discussion

Each experience datum for a robot requires a grasp execution of the robot arm, which frequently takes tens of seconds. Deep reinforcement learning-based robot grasping operations lack direct training data, and feedback from interacting with the environment may be relatively sparse or suffer from significant delay, both of which are very unfavorable for updating network parameters. Due to the complexity of the perception and control model and the size of the network parameters, training is exceedingly challenging. It should be noted that the decision result in the simulations is superior to the real-world scenarios, even if the simulated environment is similar. Hence, to increase the accuracy of the decision-making model on a physical robot, future research will investigate ways to close the gap between simulation and the real environment.

It is important to acknowledge that despite the strong performance exhibited by the trained grasp policies, there are still limitations and challenges that require attention. One such challenge arises from the difficulty encountered by perception modules in accurately perceiving objects that are highly occluded or visually ambiguous. Moreover, the grasp policies may exhibit sensitivity to variations in object properties, such as shape, texture, or weight. Addressing these challenges and improving perception and grasp planning algorithms through further research can enhance the overall performance of the methodology.

## 6. Conclusions

This paper investigates a unique attention mechanism coupled with deep reinforcement learning architecture for robot grasping. We contend that shifting and pushing should be complementary actions for situations requiring dexterity robotic operation. Although the Q-learning approach and its variations continue to be the most widely used model for robotic grasping, these algorithms do not learn to coordinate shifting and grasping. The learning model for our proposed method undergoes simulation-based self-supervision training. Our approach exceeds the other evaluated alternatives, with a grasping success

rate of 88.9% in clutter. The experimental results demonstrate that the suggested methods can generalize to unknown objects and have a noticeable improvement in grasping efficiency, step-by-step motion time, and success rate.

**Author Contributions:** Conceptualization, T.Z. and H.M.; methodology, T.Z.; software, T.Z.; validation, T.Z.; formal analysis, H.M.; investigation, T.Z.; resources, H.M.; data curation, T.Z.; writing—original draft preparation, T.Z.; writing—review and editing, H.M. All authors have read and agreed to the published version of the manuscript.

**Funding:** This research received no external funding.

**Data Availability Statement:** The data presented in this study are available on request from the corresponding author.

**Conflicts of Interest:** The authors declare no conflict of interest.

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
