# Peer review of "Grasping Unstructured Objects with Full Convolutional Network in Clutter"

_electronics, doi:10.3390/electronics12143100_

Round 1

Reviewer 1 Report

Basically it is a good job, but I have some major and minor comments and questions:

The location of Figure 1 should be after the referring to this figure. To this figure I would have also a question: why were the two RGB-D cameras located in top and side of the environment? By the top setup do the robot not avoid the sight to the bottom parts? What is UR5 robot (I think Universal Robot, but it should be named)?

How can the ambience light affect to accuracy of grasping? Are two cameras enought for this task?

The title of 2. section is at the bottom of a page by me.

In line 102 what does "high-quality" mean?

In line instead of [17-19] please use [17]-[19], since 17-19 is a mathematical operation.

In Section 2 please propose some figures or charts about the available models in the literature.

By line 137-140 and 142-152 diffent line spacing can be seen. Please check it.

By line 171 "successfully grasps" - how can we decide whether it is successful or not? Sometimes the robot can grasp the item well at the first blick, but than during the movement suddenly the item fall or change its orientation.

By line 193 "two-finger parallel gripper" how does it work? So does it use pneumatic or electric source?

By subsection 4.1. Experimental setup: please add a photo about the real environment and also two photos, what the cameras see.

Which simulation software did You use for creating the cases in subsection 4.2?

In Figure 3 the robot gripper do not grasp any item. Please add a figure, where a grasping can be seen closely. Please add another figure from the real life.

Please give some information about the grasped item, such as weight, size, orientation, etc. I see that the shape is basically rectangle. How can the special shape, like rounded or fully round one, affect the success rate?

Reviewer 2 Report

1. On page 3, line 93-95, the sentence “However, such a strategy is inefficient as creating a database of known objects or learning the discriminant equation without a given model and experience for grasping in advance is impossible. ” is too long to express the idea clearly. The author shall rewrite to make it easier to be understand;

2. The reviewer didn’t understand what does “MDP” (line 114 of page 3) means, please clarify;

3. The sentence “Pushing and grasping motion primitives learned via deep Q-learning [26] has also been suggested, achieving an task success rate of 85% for cluttered objects after around 2k training episodes. ” (line 122-124 of page 3) contains apparent grammar problems, as well as the sentence “Moreover, he attention mechanism is utilized to raise the target’s expressiveness to gain a better shifting and grasping action to output the predicted rewards. ” between line 126-128 on page 3.

4. On page 7, line 216-218, the sentence “We execute 10 runs per test, and the threshold of the action numbers is set to 2*m per episode (m is the number of objects that must be picked per case) per episode. ” shall be double checked to confirm the unit of the describe variable.

The author shall carefully check the whole article to avoid apparent misusage of English, including space (line 16 of abstract section) between words.

Reviewer 3 Report

The authors raise a very interesting topic in the field of robotics. The authors present the Deep Q-learning platform and their method for improving effectiveness over other solutions.
In the introduction, the authors present the problem well and show the reasons why they tackled the topic and why it is important.
Then they present the current state of knowledge. They refer to the research of other teams in the subject in quite detail. Correct selection of literature.
The formal presentation of the problem does not raise any doubts.
Same as a formal representation of the authors algorithm.
The description of training and testing, however concise it may be. Positive is a description of cases in the form of a drawing.
A comparison with other approaches deserves positive attention. The comparison shows here the advantage of the results obtained by the authors.
In my opinion, the discussion should be developed, although some of these postulates were included in earlier chapters.
However, this does not invalidate my good opinion of the article.

Round 2

Reviewer 1 Report

Thanks for Your work regarding improving this paper. Most of my comments were answered, but one of them remained unclear: which power supply does the end-effector use? I mean, electronic or pneumatic way? I think, the first one was used, but please write it clearly around the Figure 3. After answering it I can accept the paper.
